## [Peer Review File · Journal of Cell Science]

Beat II and Side IV keep migrating longitudinal visceral muscle precursor cells on their substratum in *Drosophila*

Na Huang, Jaqueline C. Kinold, Niklas W. G. Weiß, Nargis Piroddi, Iris Fey and Hermann Aberle

DOI: 10.1242/jcs.264157

Editor: Richa Rikhy

Review timeline

Original submission:	21 May 2025
Editorial decision:	16 July 2025
First revision received:	17 October 2025
Editorial decision:	17 November 2025
Second revision received:	16 December 2025
Accepted:	17 December 2025

Original submission

First decision letter

MS ID#: jcs.264157

MS TITLE: Beaten path II and Sidestep IV keep migrating longitudinal visceral muscle precursor cells on their substratum in *Drosophila*

AUTHORS: Na Huang; Jaqueline Kinold; Hermann Aberle

ARTICLE TYPE: Research Article

Dear Dr Aberle,

We have now reached a decision on the above manuscript.

To see the reviewers' reports and a copy of this decision letter, please go to:

As you will see, the reviewers raise a some criticisms that prevent me from accepting the paper at this stage. They suggest, however, that a revised version might prove acceptable, if you can address their concerns. If you think that you can deal satisfactorily with the criticisms on revision, I would be pleased to see a revised manuscript. We would then return it to the reviewers.

Reviewer 1

Advance summary and potential significance to field

Knowledge about the regulation of cell migration during visceral muscle differentiation has been constantly expanded over the last years in *Drosophila*. Huang and coworkers now extend this understanding by analyzing the immunoglobulin-like proteins of the Beaten path and Sidestep protein families in the present manuscript.

By screening a collection of fluorescently tagged proteins, they identified Beat IIa and IIb (Beat II) to be expressed in migrating longitudinal visceral muscles (LVMs) and consistently the proposed

binding partner Side VI in the substrate for migrating LVMs, the trunk visceral mesoderm (TVM) cells, suggesting that Beat II and Side IV probably act in a ligand-receptor dependent manner as it is already shown for the nervous system. Accordingly, the authors analyzed the expression patterns of the genes involved, generated specific mutants, and analyzed the corresponding phenotypes, which are characterized by irregular migration patterns and abnormal cell distributions of LVMs resulting in larval midguts with fewer longitudinal muscle fibres. Furthermore, Huang et al. demonstrated in cell-cell aggregation assays that Beat IIs indeed interact with Side IV and, accordingly, ectopic expression of Side IV in tracheae attracts migrating CVM cells. In summary, the authors have extended the knowledge of cell adhesion during visceral muscle migration and differentiation and conducted logical experiments to analyse the involved proteins. These analyses are well described, nicely illustrated, clearly explained, and provide novel insights into new proteins and their roles in visceral muscle migration. I only noticed a few minor points in the manuscript that should be corrected before publication. However, there is one mayor point in the result section that needs to be urgently corrected.

Comments for the author

Mayor Point

There is one mayor and two additional points in the result section that needs to be corrected and improved. The statements and results described in Line 281-285 are not shown in the Figure and therefore the corresponding Figure hints (Line 282/Line 284) as well as the Figure Legend (Line 859 and following) are wrong. The same is true for the section from Line 294 to Line 304. I assume that there is maybe a wrong labeling in the Figure Heading?

Furthermore, the relevance of the result section from Line 305 to Line 318 for the behavior of Beat IIb expressing cells seems unclear to me. This part in the present form represents more or less a summary of known results of the mentioned authors. Please clarify.

And there is one other point at the end of the Results that should be improved. The side phenotypes in Figure 8B-C looks much stronger than the phenotypes shown in Figure 3P-T while the misexpression looks "much better" in the way of having more aligned LVMs. Can you please provide us with data of larval longitudinal muscle fibres of control and misexpression embryos?

Furthermore, Figure 8C shows so many randomly scattered myoblasts, possibly attracted by ventral tracheal branches that I wonder why the attraction only occurs near the visceral branches, and the mass of scattered myoblasts is no longer visible. Can you please try to explain this?

Minor Points

Material and Methods

Line 223: Please correct 70%...

Results

Line 331: Please correct the Figure hint to Figure 1A-F.

Line 363: Please refer also to Figure 3A-B.

Line 381: Please correct the Figure hint to Figure 3P-T.

Line 391: Please clarify "18 longitudinal fibers in total".

Discussion

Line 526-542: Following this paragraph, a possible hierarchy of the components identified so far and the classification of the newly identified factors would be desirable.

Line 558: Please correct the sentence.

Figures

Line 854: As described above.

Figure Legends

Line 859: As described above.

Line 906: Please correct dramatically to strongly.

Line 949: What does "indel" means?

Line 1058: Please skip "staining".

Line 1126: Please correct "amnio serosa".

Supp. Figure Legends

Line 1136: Please mention the arrows in Figure S1M-O.

Line 1172: Please mention the arrowheads in Figure S3.

Line 1177: Please correct "stagees".

Throughout the manuscript

Please label units consistently, either without as some Journals want or correctly with spaces. And the same applies to the Figure hints without space.

Reviewer 2

Advance summary and potential significance to field

This manuscript investigates the roles of the beat and side gene families in caudal visceral mesoderm (CVM) migration and the fusion of longitudinal visceral muscle precursors (LVMps) with circular muscle precursors in *Drosophila* embryos. The study employs a combination of mutant analysis, rescue experiments, and expression pattern characterization to argue that Beat-Side interactions guide cell migration and contribute to muscle fiber formation. While the experiments are compelling and the results potentially significant, several aspects—such as embryo staging, marker usage, image quality, and quantification—require refinement. Additional controls, more rigorous quantification, and clarification of specific mechanisms would substantially strengthen the manuscript.

In particular, while the authors have done a respectable job analyzing the mutant phenotypes associated with beat IIa, beat IIb, and side IV, the study concludes by asserting that Beat/Side interactions function to support chemoattraction. Although both the in vitro studies and ectopic expression of side IV in the trachea lead to migration defects, the CVM migration in beat and side mutants (Figure 3) appears relatively normal—for example, cells still reach the anterior-most position. The authors should clarify whether they believe redundant attraction mechanisms are at play.

Additionally, it is unclear how rescuing cell death via p35 expression in side mutants results in less misdirected migration. This point should be discussed in more detail to clarify the underlying mechanism. If Beat-Side interaction leads to misdirected migration, shouldn't this still be a problem once cell death is blocked?

Overall, this is a very interesting and well-conceived study that should be suitable for publication once the necessary revisions are made.

Comments for the author

Major Comments:

1. In general, it would be helpful for the authors to specify the criteria used for embryo staging. For instance, stage 12 is marked by the beginning of germband retraction, but in Figure 1G, the indicated stage 11 appears more consistent with stage 12-13.
2. The identification of two white nuclei within the anti- \hat{I}^2 GAL signal (suggesting binucleated cells) as CVM cells fusing with fusion-competent myoblasts is unconvincing. Could the authors label the TVM-derived FCMs with additional markers, such as Sns for the membrane? Fusion should also be quantified at stages 13-15 in both wild-type and mutant embryos for better comparison. The arrowheads in the figure should be clearly defined in the legend.
3. The quality of in situ images in Figure 2 could be improved. Why not use HLH54F transcript colocalization? It would likely provide a cleaner signal than the diffuse anti-GFP staining.
4. In Supplementary Figure 2, the authors suggest that Side-IV-GFP is expressed in TVM and later in LVM precursors and fibers. Are the arrows indicating TVM? Is the staging accurate?
5. Background GFP signal is generally high in Figure 2, including for the colocalization of Side IV.
6. Beat IIa expression has been shown to occur earlier than Beat IIb; see Sun et al., *Development* (2024), doi:10.1242/dev.202262.

7. In Figure 3, developmental stages are not labeled. The authors should also present single mutants without the deficiency allele or include RNAi experiments to confirm that the phenotype is specific to beat and side genes. Migration phenotypes are not easily visible in the current images. The manuscript suggests that CVM cells are densely connected in mutants rather than streaming—what stage does this refer to? Stage 13? What about earlier, before CVM cells divide and turn posteriorly?
8. Figure 5 presents a nice experiment, and it is encouraging to see that restoring beat-IIa function rescues the deficiency phenotype. Interestingly, the rescued embryo resembles the beat-IIb single mutant. To strengthen the result, the authors should quantify the rescue. Also, can Df(3R)ex7320 be rescued with side?
9. In Figure 6, why does p35 expression in the side mutant help CVM cells remain on track? How is the loss of Side IV function in TVM rescued by caspase inhibition in CVM? This should be discussed.
10. In Figure 7, additional controls are essential: beatIIa-GFP alone, SP-GFP-beatIIb alone, and both Beats together (in the absence of Side IV). Quantification should be moved to the main figure. In addition to aggregate size, the frequency of trans interactions should also be quantified. Were clusters/aggregates detected at similar frequencies?
11. It is interesting that misexpression of side genes alters LVMP migration, suggesting that Beat-Side interaction is essential for CVM and LVMP cell movement. Does misexpression of SideIV in mutants rescue cell death?
12. Can the authors co-localize Side IV and Beat-IIb proteins in vivo to detect trans-interactions? Perhaps using a proximity ligation assay or similar technique?
13. The data largely support the model that Beats and Sides keep CVM cells on track before stage 13 and then direct dorsal-ventral migration of LVMP cells during stages 14-15. After stage 15, both genes are expressed in LVMP and fibers. Do Beat-Side interactions act in parallel to, or downstream of, FGF signaling during stages 12-13?
14. Broad side expression in the trachea does not, by itself, support a chemoattractive function for Beat/Side interaction. Where exactly is side expressed in the trachea? Line 482 hypothesizes chemoattraction, but endogenous expression is not shown (unclear in Figure 2N).
15. Line 591: It is unclear whether the reduced number of LVM fibers is directly due to apoptosis in the side mutant, e.g., in Figure 6I,J. In the rescue experiment (p35 expression in side mutant), is LVM fiber number increased? Why would rescuing apoptosis result in less misdirected migration? How was the rescue performed, and are there stage-matched, non-rescued controls?

Minor Comments:

16. Line 65: McMahon et al. 2008 is not the correct reference.
17. Line 334: The earlier expression of beat IIb at stage 10 is not shown. Since this has already been published, the authors should cite Sun et al., Development 2024.
18. Figure 8: Instead of labeling with "dsRed," presumably to detect HLH54Fb-cytoRFP, it would be clearer to label as "RFP."
19. Line 504: Prior studies have already shown that beat IIb is expressed in the CVM; these should be cited here.
20. Figure 8: Please remove the dashed lines from inside the boxes, as they obscure the view. Instead, use brackets at the sides. Arrowheads should be added to the zoomed-in panels to facilitate comparison.

Reviewer 3

Advance summary and potential significance to field

In this manuscript Aberle and colleagues are investigating the morphogenesis of the longitudinal visceral muscles in the *Drosophila* embryo. It was known that these myoblasts originate in the caudal visceral mesoderm (CVM) at the posterior end of the embryo, then they migrate towards the anterior along the forming gut epithelium, specifically above the trunk visceral mesoderm (TVM), forming the circular visceral muscles, and finally cover their surface. How this migration is regulated molecularly was poorly described.

The authors find that the 2 transmembrane/GPI anchored protein families Beat II and Side IV are specifically only expressed in the CVM and TVM, respectively. Mutants in either have similar phenotypes, resulting in defects of CVM migration and spreading. In both mutants, the CV myoblasts do not properly interact with the TVM substrate during their migration and spreading, resulting in a larger distance and in increased cell death of the CV myoblasts. This causes gaps of the longitudinal muscles over the gut, affecting the gut shape at stage 17 embryos and fewer but thicker longitudinal muscles are present over the gut in larval stages.

The authors hypothesise that Beat II interacts with Side IV, in analogy to a ligand-receptor pair. Hence, the Side IV expressing TV Myoblasts guide the Beat II expressing CV Myoblasts. This is supported by S2 cell culture data suggesting that both proteins bind to each other and by the ectopic recruitment of CV myoblasts to ectopic sites of Side IV expression along the trachea. Overall, the data are well presented, novel and convincing. I have only minor suggestions to improve the manuscript.

Comments for the author

Major comments [Please request additional experiments only if they are essential for supporting the conclusions; authors should be encouraged to highlight any claims that are preliminary or speculative, or to discuss any pitfalls or alternative interpretations in a 'Limitations' section]

Minor comments

1. I assume, there is a mis-labelling of Figure 1A-F, as according to the text, this should show Beat IIb-GFP and not HLH54F-RFP?
2. In line 345, the authors cite "data not shown". These data should be shown or the statement should be deleted.
3. GAL4 should be in capital letters.
4. The labelling in Figure 6M is hard to read.
5. The ectopic recruitment of the LVMPs by Side IV expressed in the trachea would be more convincing if it would be quantified.

First revisionAuthor response to reviewers' comments**Reviewer 1 Comments to Authors:**

Mayor Point

There is one mayor and two additional points in the result section that needs to be corrected and improved. The statements and results described in Line 281-285 are not shown in the Figure and therefore the corresponding Figure hints (Line 282/Line 284) as well as the Figure Legend (Line 859 and following) are wrong. The same is true for the section from Line 294 to Line 304. I assume that there is maybe a wrong labeling in the Figure Heading?

Authors response: We thank this reviewer for noticing our mistake. Unexpectedly, we uploaded an entirely wrong figure! We have now corrected this and show the expression pattern of Beat IIb::GFP (previously lines 282-284, lines 294-304, line 859ff). We have now replaced this figure and added the original figure (now Fig. 1). The new figure is described in lines 135-177, with minimal changes to the text (highlighted in yellow).

Furthermore, the relevance of the result section from Line 305 to Line 318 for the behavior of Beat IIb expressing cells seems unclear to me. This part in the present form represents more or less a summary of known results of the mentioned authors. Please clarify.

Authors response: We agree that this part could be omitted or shortened. However, since we examine the fusion process using specific tools and markers later in the manuscript, we thought we should introduce it here. Since beat II double mutants and side IV single mutants have fewer longitudinal fibres we wanted to present evidence that the fusion process is not affected.

And there is one other point at the end of the Results that should be improved. The side phenotypes in Figure 8B-C looks much stronger than the phenotypes shown in Figure 3P-T while the misexpression looks "much better" in the way of having more aligned LVMs. Can you please provide us with data of larval longitudinal muscle fibres of control and misexpression embryos? Furthermore, Figure 8C shows so many randomly scattered myoblasts, possibly attracted by ventral tracheal branches that I wonder why the attraction only occurs near the visceral branches, and the mass of scattered myoblasts is no longer visible. Can you please try to explain this?

Authors response: We thank this reviewer for highlighting the point. In fact, as it turned out, the pattern of longitudinal muscles in third instar larvae is quite good to evaluate. We have therefore now added additional data to Fig. 8 to show the final LVM pattern at the end of larval development (now Fig. 8H-J). As we describe in lines 384-387 ectopic expression of Side IV in a side IV mutant background caused clearly stronger phenotypes, with even fewer LVM fibres, than side IV mutants by themselves.

Minor Points

Authors response: We first would like to express our gratitude for detecting all these errors that we missed and apologize for our unthoughtfulness. Thank you very much indeed!

Material and Methods

Line 223: Please correct 70%....

Authors: This has now been corrected and reads "70% Glycerol/PBS" (Line 580).

Results

Line 331: Please correct the Figure hint to Figure 1A-F.

Authors response: This has now been corrected and reads "(Suppl. Fig. S1A-F)" (Line 191).

Line 363: Please refer also to Figure 3A-B.

Authors response: We agree and have now introduced a brief description of the domain structure of Beat II and Side in Results (Lines 221-227) and in the legend to Fig. 3A-E (lines 942-956);

Line 381: Please correct the Figure hint to Figure 3P-T.

Authors response: This has now been corrected and reads "(compare Fig.3N-Q with J-M, Suppl. Fig. 2Q-T)" (Line 246-247).

Line 391: Please clarify "18 longitudinal fibers in total".

Authors response: We revised this sentence; it now reads "...there were approximately 18 longitudinal fibers in total covering the midgut" (Line 256-257).

Discussion

Line 526-542: Following this paragraph, a possible hierarchy of the components identified so far and the classification of the newly identified factors would be desirable.

Authors response: We agree that this an interesting question. According to our observation (including dorsal views of potential midline crossing errors at early stages 11-12, we believe that Beat II and Side IV function are necessary for dorsal-ventral migration of LVMp cells (at stage 13 and later), not so much for the early migration the CVMs (stages 11-13). However, this is preliminary and needs to be further elaborated in more detail. We nevertheless add an integrating sentence in the discussion on this topic (line 432-434).

Line 558: Please correct the sentence.

Authors response: The sentence has now been corrected and reads "...expressed in R7,

additional synapses formed in Beat IIb-positive layers" (Line 447).

Figures

Line 854: As described above.

Authors response: The previous figure 1 was uploaded by mistake and has now been replaced with the original figure, please see new Fig. 1.

Figure Legends

Line 859: As described above.

Authors response: The previous figure 1 was uploaded by mistake and has now been replaced with the original figure. The legend has therefore changed only marginally, please see lines 895-920.

Line 906: Please correct dramatically to strongly.

Authors response: The sentence has now been changed to "Expression levels increase strongly during anterior..." (Line 929-930).

Line 949: What does "indel" means?

Authors response: We have now deleted the confusing word "indel". The sentence reads "...consists of deletion of 5 bp and insertion of 3 incidental base pairs during the repair process, resulting in a frame shift that causes a premature stop codon (see Methods)" (Line 949-951).

Line 1058: Please skip "staining".

Authors response: The word "staining" was now deleted. The description now reads "...stained with anti-bGal (magenta)" (Line 1024-1025).

Line 1126: Please correct "amnio serosa".

Authors response: We have now changed "amnio serosa" into "amnioserosa" (please see legend to Suppl. Fig. S1).

Supp. Figure Legends

Line 1136: Please mention the arrows in Figure S1M-O.

Authors response: We now refer to the arrows as follows "...expressed in the TVM from stage 12 onward (arrows in M-O). (please see legend to Suppl. Fig. S1).

Line 1172: Please mention the arrowheads in Figure S3.

Authors response: To refer to the arrowheads we now included the sentence "Arrowheads highlight nuclei" (please see legend to Suppl. Fig. S3).

Line 1177: Please correct "stagees".

Authors response: This has now been corrected "stages" (please see legend to Suppl. Fig. S3).

Throughout the manuscript

Please label units consistently, either without as some Journals want or correctly with spaces. And the same applies to the Figure hints without space.

Authors response: Thanks for noticing these inconsistencies. We have now introduced spaces between number and units, and within figure hints.

Reviewer 2 Comments to Authors:

Major Comments:

1. In general, it would be helpful for the authors to specify the criteria used for embryo staging. For instance, stage 12 is marked by the beginning of germband retraction, but in Figure 1G, the indicated stage 11 appears more consistent with stage 12-13.

Authors response: We thank this reviewer for raising the point. After raising the brightness of previous Fig. 1G, we noticed that the embryo was falsely assigned to stage 11. In fact it is mid stage 12, which we now indicate in Fig. 1G. We also corrected the stages in Suppl. Fig. S1A and O. In general, the embryos were staged according to Campos-Ortega and Hartenstein "The embryonic development of Drosophila melanogaster". We have now included this reference in Materials and Methods (line 550-551).

2. The identification of two white nuclei within the anti-BGAL signal (suggesting binucleated cells) as CVM cells fusing with fusion-competent myoblasts is unconvincing. Could the authors label the TVM-derived FCMs with additional markers, such as *Sns* for the membrane? Fusion should also be quantified at stages 13-15 in both wild-type and mutant embryos for better comparison. The arrowheads in the figure should be clearly defined in the legend.

Authors response: Thank you very much for suggesting this idea. Since the anti-*Sns* antibody from Susan Abmayr is no longer available, we received a new antibody and an GFP knockin line (into the *sns* locus) from Tobias Hermle (Freiburg). Unfortunately, the antibody works only after heat fixation but the fixation method destroyed the epitope for the anti-beta-Gal antibody that prevented us from performing the co-staining. We crossed therefore the *Sns::GFP* line in the background of *HLHFB- LacZ* in wild-type and side IV mutant embryos. The co-staining nicely showed that a subset of *Sns::GFP* punctae perfectly overlap with anti-beta-Gal stained LVMps at stage 15 in single confocal planes (1 AU). It appeared to us that *Sns::GFP* became quickly undetectable after fusion and did not accumulate in membranes or the cytoplasm of LVMps after fusion. For this reason and for the dense packing of CVM cells in side IV mutants at stage 13 we omitted the quantification. We included the co-localization data in Suppl. Fig. S3M-N".

3. The quality of in situ images in Figure 2 could be improved. Why not use HLH54F transcript colocalization? It would likely provide a cleaner signal than the diffuse anti-GFP staining.

Authors response: We agree that images in Fig. 2A-J' show a somewhat blurry and punctate anti-GFP staining. First, our GFP antibody generally produces somewhat punctuate backgrounds. So far we were not really successful in solving this problem (see our latest attempts with *Sns::GFP* (Suppl. Fig. 3M' and N')), which show similar problems. Second, in Fig. 2 we had to perform the in situ hybridization first, followed by anti-GFP. The high temperatures used during hybridization and washing might partially affect tissue integrity or epitope availability. As we do not have a HLH54F antisense probe at hand, we decided to keep the original images, which hopefully show convincing co-localization during stages 13-15.

4. In Supplementary Figure 2, the authors suggest that Side-IV-GFP is expressed in TVM and later in LVM precursors and fibers. Are the arrows indicating TVM? Is the staging accurate?

Authors response: We do not see data for Side IV-GFP in Suppl. Fig. S2 and therefore refer to Suppl. Fig. S1 in the following: The arrows in Suppl. Fig. S1M-O indeed mark the TVM (*FasIII*). We apologize for not mentioning this in the legend, but have now added this information, please see legend to Suppl. Fig. S1. In addition, we also found that the staging was not correct for the embryo shown in panel "O". However, since the Side IV::GFP line was lost in our laboratory, we could not repeat the staining. In addition, we could not find images of this experiment showing stage 14 embryos. Since the embryo shown in panel "O" has not reached yet stage 14, we labelled it stage 13-14.

5. Background GFP signal is generally high in Figure 2, including for the colocalization of Side IV.

Authors response: In this particular experiment (Fig. 2K-N'), we had to perform the in situ hybridization first, followed by anti-*FasIII* antibody staining. The conditions for the in situ seemed to be too harsh, as we failed several times to get *FasIII*-specific signals. For this reason, we switched to anti-GFP antibodies in a *bap > mCD8GFP* background, which worked well for younger stages (st 11-13) but older stages produced high backgrounds. Despite several trials, the background could not be eliminated.

6. Beat IIa expression has been shown to occur earlier than Beat IIb; see Sun et al., Development (2024), doi:10.1242/dev.202262.

Authors response: Thanks for noticing this important detail. We have now included the reference for Sun et al., 2024 and note that these studies are based on in situ hybridization probes, while we include additional data based on an endogenous *Beat IIb::GFP* fusion protein (see Fig. 1A and lines 193-195 in the text).

7. In Figure 3, developmental stages are not labeled. The authors should also present single mutants without the deficiency allele or include RNAi experiments to confirm that the phenotype is specific to beat and side genes. Migration phenotypes are not easily visible in the current images. The manuscript suggests that CVM cells are densely connected in mutants rather than streaming—what stage does this refer to? Stage 13? What about earlier, before CVM cells divide and turn posteriorly?

Authors response: We have now added the developmental stages to Fig. 3F-I. In addition, we also present single mutant phenotypes with and without deficiency (Suppl. Fig. S2). beat IIa and beat IIb mutants are shown over deficiency, and the respective double mutant without deficiency as is the side IV single mutant (Suppl. Fig. S2). Densely connected CVM cells in the mutants refer to stage 13, at the end of their anterior migration. We labelled this phenotype in Fig. 3F, J, N and Suppl. Fig. S2M and Q with arrows. Migratory phenotypes at earlier stage were not quite obvious to us. We also imaged dorsal views at stages 11-12 but could not find any obvious phenotypes such as midline crossing. Migration phenotypes seem to commence with the onset of dorsoventral migration.

8. Figure 5 presents a nice experiment, and it is encouraging to see that restoring beat-IIa function rescues the deficiency phenotype. Interestingly, the rescued embryo resembles the beat-IIb single mutant. To strengthen the result, the authors should quantify the rescue. Also, can Df(3R)ex7320 be rescued with side?

Authors response: Due to the dense and at times chaotic cell pattern, we quantified mutant phenotypes at later stages, in third instar larvae, as this seemed a more feasible approach. However, since the deficiency is homozygous lethal, this had to be performed in a transheterozygous background. As shown in new Fig. 5I-M, Beat IIa could partially restore the number of LVM fibres in deficiency over beat II double mutants. We included the results of the quantification in the text (line 280-286). We failed, however, to rescue with UAS-Beat IIb (HLH54Fd-GAL4) and UAS-Side IV (bap3-GAL4).

9. In Figure 6, why does p35 expression in the side mutant help CVM cells remain on track? How is the loss of Side IV function in TVM rescued by caspase inhibition in CVM? This should be discussed.

Authors response: This is a good question, we were wondering ourselves. We repeated the experiment and quantified the resulting phenotypes in LVM fibres in third instar larvae (Fig. 6M-O and Q-R). Overexpression of p35 using HLH54Fd-Gal4 indeed improved the number of LVM fibres in side IV mutant backgrounds (Fig. 6O and Q). We included these results in the text (line 326-329) and in the Discussion (line 436-439).

10. In Figure 7, additional controls are essential: beatIIa-GFP alone, SP-GFP-beatIIb alone, and both Beats together (in the absence of Side IV). Quantification should be moved to the main figure. In addition to aggregate size, the frequency of trans interactions should also be quantified. Were clusters/aggregates detected at similar frequencies?

Authors response: We agree that these are necessary controls, which were lacking. We have now performed new experiments and also quantified the cell aggregates (Fig. 7 and new Suppl. Fig. S4).

Due to problems with transfection efficiencies, the percentage of transfected cells is generally low but cells expressing interacting proteins generally separated themselves in clusters from non-transfected cells. While Beat IIa or Beat IIb or both together did not form cell-cell aggregates, when mixed together (Suppl. Fig. S4 A-A', C-C', E-E'), the presence of Side IV induced clustering (Suppl. Fig. S4 B-B', D-D', F-F'). Side IV by itself did also not interact homophilically (Suppl. Fig. S4G-G'). Quantification confirmed these results and is now shown in Fig. 7D and E. Both aggregate size and aggregation frequencies were significantly higher in heterophilic settings. However, co-transfection of both Beat IIa and Beat IIb significantly strengthened interaction with Side IV, leading to larger aggregates at higher frequencies that either of the Beat's alone. These results have been included in the main text (line 351-357).

11. It is interesting that misexpression of side genes alters LVMp migration, suggesting that Beat-Side interaction is essential for CVM and LVMp cell movement. Does misexpression of SideIV in mutants rescue cell death?

Authors response: We thank this reviewer for highlighting this point. We have now performed more experiments and added new data to the manuscript. Misexpression of Side IV in trachea of side IV mutants strongly increased phenotypic strength in LVM fibres of third instar larvae (Fig. 8H-J). In addition, it seems to rescue cell death as misguided LVMs survive attached to trachea at least until larval stages (Fig. 8E-G). These results, including a quantification of LVM numbers and widths has now been included in the manuscript (line 381-387).

12. Can the authors co-localize Side IV and Beat-IIb proteins in vivo to detect trans-interactions? Perhaps using a proximity ligation assay or similar technique?

Authors response: This is certainly an important point but due to the loss of the Side IV::GFP line this was not attempted.

13. The data largely support the model that Beats and Sides keep CVM cells on track before stage 13 and then direct dorsal-ventral migration of LVMp cells during stages 14-15. After stage 15, both genes are expressed in LVMp and fibers. Do Beat-Side interactions act in parallel to, or downstream of, FGF signaling during stages 12-13?

Authors response: This is an important point and we just received some of the lines and constructs from Arno Müllers laboratory to examine the hierarchy between Beat-Side and the FGF pathway. As it stands, and based on our experimental observations, the main phenotype seems to be an abnormal dorsal-ventral migration starting at the end of stage 13, making a parallel interaction unlikely but Beats and Sides may well be downstream of FGF signals but this has to be validated experimentally.

14. Broad side expression in the trachea does not, by itself, support a chemoattractive function for Beat/Side interaction. Where exactly is side expressed in the trachea? Line 482 hypothesizes chemoattraction, but endogenous expression is not shown (unclear in Figure 2N).

Authors response: Based on our expression pattern analysis, endogenous side IV does not seem to be expressed in tracheal. We have, for example, side IV mRNA not detected in trachea in our in situ hybridization experiments. Even ectopic expression of Side IV in trachea in a wild-type background does not seem to be sufficient to recruit LVMs. If Side IV is lacking, such as in side IV mutants, however, forced expression of Side IV on trachea is sufficient to attract LVMs.

15. Line 591: It is unclear whether the reduced number of LVM fibers is directly due to apoptosis in the side mutant, e.g., in Figure 6I,J. In the rescue experiment (p35 expression in side mutant), is LVM fiber number increased? Why would rescuing apoptosis result in less misdirected migration? How was the rescue performed, and are there stage-matched, non- rescued controls?

Authors response: We have performed new and independent experiments and, as stated above, overexpression of p35 indeed increased LVM fibres in third instar larvae. Since we agree to this reviewer that rescuing apoptosis is not directly linked to restoring migration, we argue in the discussion that up- or downregulation of molecules with redundant functions help to compensate for the loss of side IV (lines 439-441). We also changed the last sentence in the Discussion in the same sense (previously line 591, now line 484-486).

Minor Comments:

16. Line 65: McMahon et al. 2008 is not the correct reference.

Authors response: We agree and have now replaced the reference with Azpiazu et al. 1993, Georgias et al. 1997 and Sun et al. 2020 (Line 66).

17. Line 334: The earlier expression of beat IIb at stage 10 is not shown. Since this has already been published, the authors should cite Sun et al., Development 2024.

Authors response: We agree and have now inserted Sun et al. 2024 (Line 194). We also show Beat IIb expression at stage 10 in Fig. 1A.

18. Figure 8: Instead of labeling with "dsRed," presumably to detect HLH54Fb-cytoRFP, it would be clearer to label as "RFP."

Authors response: We agree that this is less confusion and have changed now the labelling accordingly. We also define in Materials & Methods that the anti-dsRed antibody detects RFP, a monomeric form derived from dsRed (line 568-569).

19. Line 504: Prior studies have already shown that beat IIb is expressed in the CVM; these should be cited here.

Authors response: This is true and we have now included to references of earlier studies discovering the expression of beat II genes in CVM (line 394-395).

20. Figure 8: Please remove the dashed lines from inside the boxes, as they obscure the view. Instead, use brackets at the sides. Arrowheads should be added to the zoomed-in panels to facilitate comparison.

Authors response: We agree and have removed the drawings for clarity. In addition, we have re-arranged the entire figure, including labels (asterisk) for the visceral branch of the trachea and

arrows to show the tight attachment of LVMPs to these branches.

Reviewer 3 Comments to Authors:

Minor comments

1. I assume, there is a mis-labelling of Figure 1A-F, as according to the text, this should show Beat IIb-GFP and not HLH54F-RFP?

Authors response: We thank Reviewer 3 for noticing our mistake, which was caused by uploading the wrong figure. We have now exchanged this figure for the current version (now Fig. 1), which actually shows Beat IIb::GFP. There are some minor changes to the text, too, which we highlighted in yellow (lines 135-177).

2. In line 345, the authors cite "data not shown". These data should be shown or the statement should be deleted.

Authors response: We agree and have now deleted this sentence (line 204-205). The results of the side IV in situ hybridization are still shown in co-stainings with anti-GFP in TVM cells (Fig. 2K-N').

3. GAL4 should be in capital letters.

Authors response: Thanks for noticing this mistake. We have now changed "Gal4" to "GAL4" throughout the manuscript.

4. The labelling in Figure 6M is hard to read.

Authors response: We have now adjusted font size to 8 point in all figures and the labelling in Fig. 6M should now be easier to read.

5. The ectopic recruitment of the LVMPs by Side IV expressed in the trachea would be more convincing if it would be quantified.

Authors response: We agree and have substantially worked on this figure. For quantification, we now include data from third instar larvae, as their persisting LVM phenotypes are better amenable for quantification (Fig. 8K). To confirm the increase in phenotypic strength in side IV mutant larvae expressing additionally Side IV in trachea, we included also confocal micrographs showing severe LVM phenotypes (Fig. 8H-J). We also included images of whole mount first instar larvae showing that at least some LVMP cells attracted to tracheal branches during embryogenesis maintain their contact with the trachea until larval stages (see Fig. 8E-G).

Second decision letter

MS ID#: jcs.264157R1

MS TITLE: Beat II and Side IV keep migrating longitudinal visceral muscle precursor cells on their substratum in *Drosophila*

AUTHORS: Na Huang; Jaqueline Kinold; Niklas Walter Gerhard Weiß; Nargis Piroddi; Iris Fey; Hermann Aberle

ARTICLE TYPE: Research Article

Dear Dr Aberle,

We have now reached a decision on the above manuscript.

As you will see, the reviewers gave favourable reports but raised some critical points that will require amendments to your manuscript. I hope that you will be able to carry these out because I would like to be able to accept your paper based on the responses to the comments and revisions.

Reviewer 1*Advance summary and potential significance to field*

Knowledge about the regulation of cell migration during visceral muscle differentiation has been constantly expanded over the last years in *Drosophila*. Huang and coworkers now extend this understanding by analyzing the immunoglobulin-like proteins of the Beaten path and Sidestep protein families in the present manuscript.

By screening a collection of fluorescently tagged proteins, they identified Beat IIa and IIb (Beat II) to be expressed in migrating longitudinal visceral muscles (LVMs) and consistently the proposed binding partner Side IV in the substrate for migrating LVMs, the trunk visceral mesoderm (TVM) cells, suggesting that Beat II and Side IV probably act in a ligand-receptor dependent manner as it is already shown for the nervous system. Accordingly, the authors analyzed the expression patterns of the genes involved, generated specific mutants, and analyzed the corresponding phenotypes, which are characterized by irregular migration patterns and abnormal cell distributions of LVMs resulting in larval midguts with fewer longitudinal muscle fibres. Furthermore, Huang et al. demonstrated in cell-cell aggregation assays that Beat IIs indeed interact with Side IV and, accordingly, ectopic expression of Side IV in tracheae attracts migrating CVM cells.

In summary, the authors have extended the knowledge of cell adhesion during visceral muscle migration and differentiation and conducted logical experiments to analyse the involved proteins. These analyses are well described, nicely illustrated, clearly explained, and provide novel insights into new proteins and their roles for visceral muscle migration.

I am pleased that the authors have corrected all the requested points and further revised the manuscript, which has led to a significant improvement and I recommend the manuscript for publication.

Comments for the author

Major comments [Please request additional experiments only if they are essential for supporting the conclusions; authors should be encouraged to highlight any claims that are preliminary or speculative, or to discuss any pitfalls or alternative interpretations in a 'Limitations' section]

Minor comments

Line 227-230: Please clarify if you use mutants from Osaka et al. 2024 or your own mutants.

Line 930-932: Why is the strong heart expression from Beat 2a and the strong amnioserosa expression from Beat 2b not visible in the in situ hybridizations?

Reviewer 2*Advance summary and potential significance to field*

The authors have convincingly showed that Beat IIa, Beat IIb, and Side IV function to support CVM/LVMp association of migrating cells (expressing Beats) with the TVM (expressing Side IV). This is the first demonstrating of Beats/Side in supporting non-neuronal cell migration. The authors have been responsive to the reviewer's requests. There is some concern that they have lost the SideIV:GFP fly stocks but hopefully that can be re-obtained from another source (?).

Comments for the author

Major comment - correct referencing:

(1) Line 186: "From late stage 13 to 14, when LVMps develop and start to migrate dorsoventrally, beat IIa mRNA was strongly detected in these cells (Fig. 2C-C')."

It is not clear that the signal is strong at stage 13/14. In fact it looks weaker at stage 14 compared to stage 13.

(2) Line 194: The text is misrepresenting results from Sun et al. 2024 (Fig. 2) which found the opposite to what is written. They showed that *beatIIb* is earlier than *beatIIa*. Consistent with the presented results. Please revise.

Minor comments - to improve clarity:

(3) Line 188: In sup Fig S1E'F' it is not clear if signal for Beat IIa-GFP is detectable. Can you show the GFP only channel without RFP

(4) Line 208: In Figure 2K-N' please also show the side IV mRNA signal alone without the gfp signal.

(5) Suppl. Fig. S1M-R - also show anti-GFP signal alone. These data in particular are not the best evidence.

Why doesn't the side in situ show the expression at stage 15 (the embryo in Figure 1N,N' is not very convincing)

(6) Line 317: Please clarify in the text that Fig. 6C-C' is wildtype and 6F-F' are side IV mutants. Also the results seem to contradict themselves. Figure 6B show what appear to be apoptotic cells. Wouldn't these be expected to be detectable using TUNEL and thus wouldn't some low level signal be expected for experiment shown in 6C,C'?

(7) Fig 6P-R: please use consistent colors for genotypes (e.g. blue for all control samples)

(8) Fig 8C-D": it would be better to show that Side was indeed ectopically expressed through staining with anti-V5. Was this done?

(9) Fig 8G': without a reference to tracheal dorsal trunk, I cannot interpret the pattern shown. Consider adding magnified views. Could this be with the *bth*-GFP as shown in panels above but for the larval stage?

Second revision

Author response to reviewers' comments

Comments to Reviewer 1:

Minor comments

a) Line 227-230: Please clarify if you use mutants from Osaka et al. 2024 or your own mutants.

Authors response: To better clarify which mutants were used, we now provide a better reference to Suppl. Fig. S2 by highlighting the relevant panels Suppl. Fig. S2A-L (line 230). In addition, we introduced a new sentence to clarify that we created our double mutant in the background of the *beat IIa+2* allele (Line 237-239).

We also inserted the reference Osaka et al., 2024 in the corresponding paragraph in the Materials and methods section (Line 512).

b) Line 930-932: Why is the strong heart expression from Beat 2a and the strong amnioserosa expression from Beat 2b not visible in the in situ hybridizations?

Authors response: We thank Reviewer 1 for highlighting these potential contradictions. We have now re-examined our images and published figures. Based on their position within the embryo and developmental stage, we indeed find that a subset of heart cells expresses *beat IIa* transcripts in our images. We also found *in situ* hybridization images in Bae et al., showing *beat IIa* expression in a few dorsal cells (Bae et al., Mech. Dev. 2017, see Fig. 3, stage 13 there) and at the expression pattern database at BDGP that resembled heart cells and our expression patterns. We have therefore decided to indicate these cells in our manuscript in Fig. 2D (arrows) and in the legend (Line 937).

For *beat IIb* mRNA, we could not find any clear evidence for amnioserosa expression in our in

situ experiments and images. This was similar in Bae et al., *Mech. Dev.* 2017 (Fig. 3), Pipes et al., *Dev.* 2001 (Fig. 3) and Sun et al., *Dev.* 2024 (Fig. 1 and 2). Interestingly, in Sun et al., *beat Ila* rather than *beat IIb* seemed to be expressed in the amnioserosa (Fig. 2, stage 13). Due to these uncertainties and also our own lack of clear evidence, we decided that more experiments are need for a clear conclusion.

Comments to Reviewer 2:

Major comment - correct referencing:

(1) Line 186: "From late stage 13 to 14, when LVMps develop and start to migrate dorsoventrally, beat Ila mRNA was strongly detected in these cells (Fig. 2C-C')." It is not clear that the signal is strong at stage 13/14. In fact it looks weaker at stage 14 compared to stage 13.

Authors response: We thank Reviewer 2 for carefully comparing text and figures. We now went through all our images and found that expression of *beat Ila* is highest at stage 13. We therefore agree to this Reviewers observation and changed the text accordingly. We now write that "the signal increased in migrating CVM cells and was strongest at stage stage 13 (Fig. 2B-B'" (Line185).

In addition, we state that "...migrate dorsally, *beat Ila* mRNA expression commenced to decline (Fig. 2C-C'" (Line 187).

(2) Line 194: The text is misrepresenting results from Sun et al. 2024 (Fig. 2) which found the opposite to what is written. They showed that *beatIIb* is earlier than *beatIla*. Consistent with the presented results. Please revise.

Authors response: Thanks for correcting this mistake. We were in fact confused by the magenta colour and the asterisks in that figure (Fig. 2; Sun et al. 2024). We have now revised the sentence to state that we confirm the independent observation: ", thereby confirming prior observations using probes for in situ hybridizations (Sun et al., 2024)." (Line 194).

Minor comments - to improve clarity:

(3) Line 188: In sup Fig S1E'F' it is not clear if signal for Beat Ila-GFP is detectable. Can you show the GFP only channel without RFP

Authors response: We agree that the HLH54Fb-cytoRFP signals in Suppl. Fig. S1E',F' shown in magenta overlay somewhat weaker *BeatIla::GFP* signals (green). For better visibility of these *BeatIla::GFP* signals, we have therefore created a new supplementary figure showing the GFP signals in single channel views. Suppl. Fig. S5K-N' now shows these images for Suppl. Fig. S1C-F'. Individual longitudinal muscle fibres are marked by green arrowheads. We also refer readers in the legend to Suppl. Fig. S1 to these single channel images in Suppl. Fig. S5 (Line 42 in supplemental materials).

(4) Line 208: In Figure 2K-N' please also show the side IV mRNA signal alone without the gfp signal. Authors response: In accordance with (3) above, we have now added single channel views of Fig. 2K-N' showing only *side IV* mRNA expression in Suppl. Fig. S5A-D'. We also refer to these images in the manuscript: "(see single channel images in Suppl. Fig. S5A-D')." (Line 210).

(5) Suppl. Fig. S1M-R - also show anti-GFP signal alone. These data in particular are not the best evidence.

Why doesn't the side in situ show the expression at stage 15 (the embryo in Figure 1N,N' is not very convincing)

Authors response: To avoid confusion, we assume that Reviewer 2 refers here to Fig. 2N, N' showing *side IV* mRNA expression (Fig. 1N is a scheme).

We now provide the requested single channel images derived from Suppl. Fig. S1M-R showing the expression of *Side IV::GFP* only. We added these images to Suppl. Fig. S5E-J. Developing longitudinal visceral muscle fibres (LVMps and LVMs) showing *Side-IV::GFP* expression have been labelled with green arrowheads at stage 15-17.

We also refer to these images in the text: "..., see also single channel images in Suppl. Fig. S5E-J for clarity)" (Line 217).

(6) Line 317: Please clarify in the text that Fig. 6C-C' is wildtype and 6F-F' are side IV mutants. Also the results seem to contradict themselves. Figure 6B show what appear to be apoptotic cells. Wouldn't these be expected to be detectable using TUNEL and thus wouldn't some low level signal be expected for experiment shown in 6C,C'?

Authors response: Thanks for this note. We have now added "in *side IV* mutants" to clarify the genotypes (Line 320).

In addition, we went through our images and found evidence for occasional apoptosis among migrating CVM cells in control embryos. To reflect this finding, we have changed the images in Fig. 6C,C' and changed the text accordingly to indicate this: "..., but only a few specific signals..." (Line 318).

(7) Fig 6P-R: please use consistent colors for genotypes (e.g. blue for all control samples)

Authors response: We thank this reviewer for detecting this inconsistency and apologize for this unnecessary mistake. We have now adjusted the colours in these graphs accordingly, and each genotype is now depicted in the same colour (see Fig. 6P-R). We also changed the lay-out of the figure slightly. There was no necessity to correct the legend to Fig. 6P-R.

(8) Fig 8C-D": it would be better to show that Side was indeed ectopically expressed through staining with anti-V5. Was this done?

Authors response: Yes, this was done. Expression of Side IV-V5 on trachea is shown in Fig. 8C', D' in turquoise colour (we chose a different colour to distinguish the V5 epitope from GFP above (Fig. 8A', B')).

(9) Fig 8G': without a reference to tracheal dorsal trunk, I cannot interpret the pattern shown. Consider adding magnified views. Could this be with the bth-GFP as shown in panels above but for the larval stage?

Authors response: Thanks for this suggestion, which we adopt accordingly. We have now added magnified views to better highlight the tracheal trunk. To better label the extend of the dorsal trunk, we replaced the white arrows (previously in Fig. 8E-G) with green broken lines that follow the tracheal trunk over its entire length (new Fig. 8E-G').

In addition, we show a *side IV* mutant L1 larvae in Fig. 8F" overexpressing mCD8GFP in trachea that reveals the entire dorsal tracheal trunk (green, white arrows) and LVMs (magenta).

Comments to Reviewer 3:

I want to congratulate the authors on this very interesting and well executed study.

Authors response: Thank you very much for your kind response and help.

Third decision letter

MS ID#: jcs.264157R2

MS Title: Beat II and Side IV keep migrating longitudinal visceral muscle precursor cells on their substratum in *Drosophila*

Authors: Na Huang; Jaqueline Carolin Kinold; Niklas Walter Gerhard Weiß; Nargis Piroddi; Iris Fey; Hermann Aberle

Article Type: Research Article

Dear Dr Aberle,

I am happy to tell you that your manuscript has been accepted for publication in Journal of Cell Science, pending standard publication integrity checks.